# Non-zero mean alpha oscillations revealed with computational model and empirical data

**Alina A. Studenova** [1,2]*, **Arno Villringer** [1,3], **Vadim V. Nikulin** [1,2,4,5]

**1** Department of Neurology, Max Planck Institute for Human Cognitive and Brain Sciences, Leipzig, Germany, **2** Institute for Cognitive Neuroscience, National Research University Higher School of Economics, Moscow, Russia, **3** Department of Cognitive Neurology, University Hospital Leipzig, Leipzig, Germany, **4** Neurophysics Group, Department of Neurology, Charité-Universitätsmedizin Berlin, Berlin, Germany, **5** Bernstein Center for Computational Neuroscience, Berlin, Germany

* studenova@cbs.mpg.de

**Data Availability Statement:** The EEG data were recorded as part of the "Leipzig Cohort for Mind-Body-Emotion Interactions" data set (LEMON). Data are publicly available at http://fcon_1000. projects.nitrc.org/indi/retro/MPI_LEMON.html.

## Abstract

Ongoing oscillations and evoked responses are two main types of neuronal activity obtained with diverse electrophysiological recordings (EEG/MEG/iEEG/LFP). Although typically studied separately, they might in fact be closely related. One possibility to unite them is to demonstrate that neuronal oscillations have non-zero mean which predicts that stimulus- or task-triggered amplitude modulation of oscillations can contribute to the generation of evoked responses. We validated this mechanism using computational modelling and analysis of a large EEG data set. With a biophysical model, we indeed demonstrated that intracellular currents in the neuron are asymmetric and, consequently, the mean of alpha oscillations is non-zero. To understand the effect that neuronal currents exert on oscillatory mean, we varied several biophysical and morphological properties of neurons in the network, such as voltage-gated channel densities, length of dendrites, and intensity of incoming stimuli. For a very large range of model parameters, we observed evidence for non-zero mean of oscillations. Complimentary, we analysed empirical rest EEG recordings of 90 participants (50 young, 40 elderly) and, with spatio-spectral decomposition, detected at least one spatially-filtered oscillatory component of non-zero mean alpha oscillations in 93% of participants. In order to explain a complex relationship between the dynamics of amplitude-envelope and corresponding baseline shifts, we performed additional simulations with simple oscillators coupled with different time delays. We demonstrated that the extent of spatial synchronisation may obscure macroscopic estimation of alpha rhythm modulation while leaving baseline shifts unchanged. Overall, our results predict that amplitude modulation of neural oscillations should at least partially explain the generation of evoked responses. Therefore, inference about changes in evoked responses with respect to cognitive conditions, age or neuropathologies should be constructed while taking into account oscillatory neuronal dynamics.

**Funding:** VVN was supported in part by Deutsche Forschungsgemeinschaft (German Research Foundation) Project ID 424778381 TRR 295. The funders had no role in study design, data collection and analysis, decision to publish, or preparation of the manuscript.

**Competing interests:** The authors have declared that no competing interests exist.

## Author summary

A large part of our knowledge about functioning of the human brain is based on non-invasive assessment of neuronal processing with electro- and magnetoencephalography (EEG/MEG). Interestingly, all types of neural processing can be captured with just two types of recorded events: evoked responses and oscillations. Evoked response is a reaction occurring to any auditory, visual or motor event. Oscillations are rhythmic activity that is spontaneous, i.e. is present all the time and may or may not be related to the stimulus. While typically studied separately, evoked responses and oscillations might be related—the change in the amplitude of oscillations could lead to an evoked response. However, it is true only for oscillations with non-zero mean. For oscillations, having a non-zero mean implies that averaged values of the upper and lower half of the oscillatory wave are not equal. In our study, we show that the most prominent rhythm in the human brain—the alpha rhythm—has non-zero mean. This, in turn, implies that many evoked responses can indeed be understood via an amplitude modulation of non-zero mean oscillations. Consecutively, such link opens new perspectives for the interpretation of results from studies investigating sensory, motor and cognitive processes.

## Introduction

Neuronal oscillations represent one of the main forms of neuronal activity in the human brain and they have been implicated in diverse sensory, motor and cognitive functions [1]. Another prominent form of neuronal activity is evoked responses (evoked potentials, evoked fields) which are transient phase-locked events produced by stimuli, movement and cognitive operations. These two phenomena are usually studied separately but there are two general ways through which they can be interrelated. On the one hand, ongoing neuronal oscillations can modulate evoked responses [2–5]. These modulations are probably due to the fact that neuronal oscillations, such as alpha and beta rhythms, are associated with changes in cortical excitability which in turn affect the recruitment of neurons responding to stimuli. On the other hand, evoked responses may, in fact, reflect changes in certain aspects of neuronal oscillations from which it would follow that both phenomena may be closely linked and share a neurophysiological origin.

Three mechanisms of evoked responses have been proposed over several decades: the additive mechanism [6–9], the phase reset mechanism [10–13], and the baseline-shift mechanism (BSM) [4, 14–16]. The additive mechanism states that evoked responses are generated independently of the ongoing oscillations, that is to say, "in addition to" (i.e. evoked response is a stand-alone phenomenon). The phase reset mechanism suggests that after a stimulus presentation phases of ongoing oscillators switch to a particular value (i.e., evoked responses are oscillations). Finally, BSM implies that any amplitude modulation of ongoing oscillations with a non-zero oscillatory mean (OM) leads to the generation of evoked responses (i.e. posits that evoked responses and oscillations reflect the same process). Note that the BSM scenario supposes non-phase-locked oscillations with non-zero OM; when averaging over trials, opposite phases of oscillations cancel out and evoked response appears in the cumulative signal. There is an ongoing debate in the literature on whether auditory [7, 8, 17], visual [4, 9, 11–13, 17], and somatosensory [14] evoked responses can be generated through the additive, phase reset mechanism, or BSM, and currently, the evidence is inconclusive for some evoked responses and completely absent for others. It should be noted that these mechanisms are not assumed to be mutually exclusive and may co-exist. Moreover, they may manifest at different latencies

of evoked-response generation and in different conditions. The third mechanism—BSM—was introduced relatively recently and so far was underexplored in studies and computational models. In the current study, we reinforced the theory behind BSM.

BSM gives rise to the following predictions: 1) post-stimulus oscillations are accompanied by amplitude modulation, 2) non-zero OM sign and direction of amplitude modulation (increase or decrease) determine the polarity of an evoked response, 3) time and spatial distribution of modulation correlate with both the evoked response time course and spatial topography. Consistent with the predictions, the presence of non-zero OM has been previously directly confirmed [14, 16] and the generation of evoked responses has been found to be consistent with the idea of BSM [4, 15, 18]. Moreover, many empirical studies assessed cognitive functioning with simultaneous analysis of evoked responses and oscillations, showing consistent correlation of evoked response amplitude and/or latency with the modulation in oscillations [19–23]. However, several other studies observed little or no correlation [24, 25]. Fukuda et al. [24] found counter evidence for alpha modulation and evoked response relatedness in the visual working memory task. The main finding was that the strength of alpha amplitude and amplitude of evoked response were not correlated between different memory array sizes. Xia et al. [25] examined evoked response P600 and suppression of alpha activity in the verbal memory task in healthy elderly participants and in patients with mild cognitive impairment and Alzheimer's disease. The study showed two phenomena to be distinct based on the fact that they lacked correlation on a single-trial level and acted as complementary metrics in multiple regression while predicting California Verbal Learning Test scores. In the current study, we considered factors that might have caused conflicting results. We assumed that, possibly, not every alpha source presents detectable baseline shifts on the level of EEG/MEG, and that the estimation of baseline shifts may be complicated in the elderly or clinical population (for instance, due to the low power of oscillations [26]).

The intuition about oscillations having non-zero OM stems from asymmetrical morphological and biophysical properties of neurons and neuronal networks that may have an effect on the generation of oscillations in a way that creates a non-zero OM activity. These asymmetries emerge on several scales: a membrane, a cell, and external inputs. On the membrane level, inward and outward currents flow through synapses and voltage- and ligand-activated channels that are placed asymmetrically along the neuron [27–30]. The placement of synapses over the dendrites of one cell is biased in a way that excitatory synapses are clustered on spines of apical and basal dendrites, and inhibitory synapses are mostly located on dendritic shafts, soma and axon initial segment [29, 30]. In addition to synapses, charges flow through voltage- and ligand-gated channels that are scattered along the membrane across all parts of dendrites and soma [27, 28]. On the cellular level, the contribution of various dendrites to a current dipole is not equal and depends on the orientation of a process with respect to the longitudinal axis of the apical dendrite [31, 32]. For instance, oblique dendrites, which are oriented in parallel to the cortical surface and transversely to the main axis of a neuron, have little effect on the net dipole. As for the basal dendrites, they are oriented in different directions (schematic representation on Fig 1A), meaning the contribution of primary currents in basal dendrites is scaled to some degree [32]. On the exogenous inputs level, proximal and distal drives arrive at different layers of the neocortex, thus creating an asymmetrical redistribution of charges [33, 34]. In particular, feedforward thalamocortical drive arrives predominantly to basal dendrites of pyramidal neurons, whereas feedback connections—to apical dendrites. The strength of thalamocortical and corticocortical connections may change irrespectively from each other, thus creating asymmetries in currents. Overall, based on theoretical assumptions about neuronal currents, it is highly unlikely that currents flowing towards soma will have the same magnitude as outward dendritic currents [14, 15]. However, as previous research on BSM was focused on

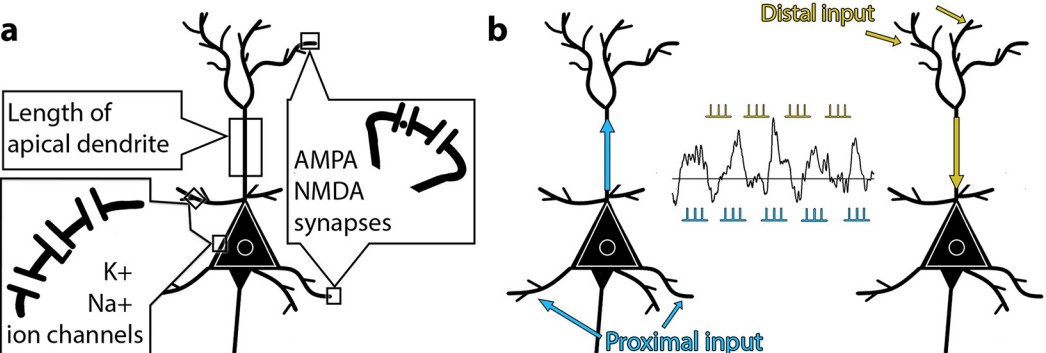

**Fig 1. Schematic representation of a pyramidal neuron as it was modelled in Human Neocortical Neurosolver (HNN)** [32]. **a**. Parameters of the model that were subjected to change: strength of incoming inputs via synaptic weights on AMPA and NMDA synaptic receptors, the density of voltage-gated sodium and potassium ion channels, and the length of apical dendrites. **b**. A schematic illustration of the effect that incoming inputs exert on the dendritic currents. The proximal drive is a simulation of thalamic activity coming to the granular layer of the cortex, which is later relayed to layer II/III and layer V. Distal drive is excitatory input from non-lemniscal thalamic sources and/or other cortical areas. Proximal connections terminate on the basal dendrites of pyramidal neurons, while distal input is reaching apical dendrites. Therefore, currents that are generated in the neuron as a response to proximal stimulation flow predominantly in the upward direction (with respect to the surface of the cortex), and distal drive creates downward currents [32]. Spontaneous alpha rhythm emerges when the delay between proximal and distal inputs is 50 ms, and both are delivered to the network with a frequency of 10 Hz [32, 40, 41]. However, simultaneous presence of both proximal and distal inputs is not essential for the emergence of the alpha rhythm (see Results/Asymmetric currents lead to non-zero mean oscillations). The current figure demonstrates the most biologically plausible layout of inputs' allocation.

EEG/MEG findings, we checked the aforementioned theoretical statements with a computational model. We supposed that the change in parameters of the model would bring about the change in cellular currents. Subsequently, when currents become more or less balanced, the mean of oscillations would change.

Despite that, there is no strict necessity to measure OM directly. If neuronal oscillations have a non-zero OM, any modulation of oscillations' amplitude affects the mean as well, thus leading to a baseline shift in lower frequency [14, 16]. Consequently, on the macroscopic recordings, there would be an association between low-frequency time course and the amplitude envelope of ongoing oscillations (see Eq 1 in the Methods section). This association is present in both stimulus-based and rest-state data because alpha rhythm modulation is present during both active and resting states. Therefore, for BSM, amplitude modulation bears a significant role both for observation of evoked responses and for empirical estimation of OM. However, the prominence of amplitude modulation may change. Therefore, as previous research sampled data exclusively from young adults and only during either eyes-closed or eyes-open state [4, 14, 15], and the sample size was relatively small, we validated BSM on the bigger data set with both eyes-closed and eyes-open conditions and with two age groups. We expected to see a manifestation of baseline shifts in the majority of participants but also hypothesised that elderly participants may deviate from young participants in the manifestation of baseline shifts. A decrease of spectral alpha power is a prominent feature of the ageing brain [26], and this issue may obscure the identification of a correspondence between alpha envelope and low-frequency baseline shifts.

In the present study, we aimed to verify the prediction about asymmetric currents leading to non-zero OM with the use of a biophysically realistic computational model. We examined a model of a small cortical patch that generates alpha oscillations—Human Neocortical Neurosolver (HNN) [32]. To understand how OM can be affected by diverse changes in the model on the level of a single neuron, we varied several biophysical and morphological properties of

neurons in the network, such as voltage-gated channel densities, length of dendrites, and intensity of incoming stimuli. Next, we attempted to validate BSM further on the large data set including young and elderly participants with eyes-open and eyes-closed sessions. Finally, we elaborated on the factors that might mask a correspondence between the amplitude modulation of oscillations and corresponding baseline shifts. For this, we investigated the effects of spatial synchronisation within a network with population modelling using simple oscillators [35].

## Materials and methods

The main prerequisites for BSM are 1) the non-zero OM and 2) amplitude modulation of oscillations [14]. This mechanism can be summarised with the equation:

$$x = \sum_{i=1}^{M} S_i$$

$$S_i = A(t)[\cos(2\pi ft + \theta) + r] = A(t)\cos(2\pi ft + \theta) + A(t)r$$

(1)

where $S_i$—data from a single oscillator,
$f$—some arbitrary frequency of oscillations in the population,
$\theta$—some arbitrary phase,
$r$—non-zero OM,
$A(t)$—amplitude modulation,
$A(t)r$—baseline shift that accompanies oscillations,
$M$—number of oscillators.

When averaging is performed, phases of oscillations across trials cancel each other, but the baseline shifts persist and emerge on a summed signal. The shift may have a positive or negative direction depending on the orientation of a dipole and on the sign of an underlying OM [4, 14, 15].

In the current study, we adopted an integral strategy and attempted to demonstrate the following: 1) asymmetric currents lead to a non-zero OM in alpha oscillations ($r$ term in Eq 1 is not equal to zero); 2) if oscillations possess the property of a non-zero OM, modulation affects OM as well, thus leading to the emergence of a baseline shift in lower frequency ($A(t)$ term is present both in amplitude envelope and in low-frequency component); 3) the prominence of amplitude modulation may change due to spatial synchronisation and, therefore, affect the relationship between amplitude modulation and corresponding baseline-shifts ($A(t)$ in $A(t)\cos(2\pi ft + \theta)$ may not be evident in EEG/MEG data).

### Simulation of the alpha rhythm with a biophysical model

In this study, a biologically plausible model of neuronal network generating alpha rhythm was selected to explore a non-zero property of oscillations. Human Neocortical Neurosolver (HNN), a software tool and an underlying model was developed by Stephanie R. Jones laboratory in Brown University [32] for investigation of cellular mechanisms of both ongoing and evoked activity. Here we provide a brief description of neuronal architecture and principles behind the generation of alpha rhythm as it was built in HNN. For a more detailed report, please refer to [32, 36–39].

The model consists of pyramidal neurons and interneurons in layer II/III and layer V of the neocortex, with the ratio of pyramidal neurons to interneurons as three to one and a total count of 270 cells. Each pyramidal cell is modelled as having a soma and a dendritic tree that consists of several compartments [38]. Interneurons are known to contribute significantly less

to the potential/ field; hence they are modelled as a single somatic compartment cell. For each compartment, membrane voltage is computed from the standard Hodgkin-Huxley equations, and current between compartments flows according to cable theory [32]. Biophysically realistic flow of ions inside the neuron is achieved by simulating several active currents: a fast sodium current, a delayed rectifier potassium current, an adapting potassium current, and a leak current in pyramidal neurons in layer II/III, same currents and a calcium current, a potassium-activated calcium current, a low-threshold calcium current and a hyperpolarization-activated mixed cation current in pyramidal neurons in layer V [37, 38]. The strength of currents can be adjusted in the interface by setting the channel density for each current. In addition to active currents, each cell receives excitatory and inhibitory input from the neighbours. Local dynamics in the network are realised via AMPA, NMDA and GABAa, b synaptic receptors. Several exogenous drives are available for investigation—evoked inputs, rhythmic inputs, Poisson and tonic drives. Rhythmic inputs are of the most interest for us because they initiate ongoing oscillations. Rhythmic drive is simulated as spike trains coming from "outside", it is excitatory and realised through AMPA and NMDA synaptic receptors [32]. These rhythmic inputs come from two directions—proximal and distal. Proximal drive represents thalamic stimulation, and it arrives at both layer II/III and layer V pyramidal neurons, with the latter connection being delayed (0.1 ms to layer II/III and 1.0 ms to layer V). Proximal input connections contact basal dendrites of pyramidal cells and hence generate current flowing in the apical dendrite in an upward direction. The distal drive is a stimulation coming from other cortical areas and non-lemniscal thalamic sources. It reaches apical dendrites of pyramidal neurons in both cortical layers simultaneously and induces current flowing down the apical dendrite [38]. The strength of inputs is controlled with the weights of the AMPA and NMDA receptors, and the number of bursts in the spike train. Spontaneous alpha rhythm emerges when the delay between proximal and distal input is exactly 50 ms, and they are delivered to the network with a frequency of 10 Hz [32, 40, 41]. The main output of the simulation is a current dipole estimate. The current dipole is computed as the sum of the intracellular currents within the dendrites projected onto the direction parallel to the apical dendrite plane. The current dipole quantifies units of current per unit of distance, therefore its measurement unit is Am (Ampere x meter) [32].

To investigate the effect of currents on OM, we varied several parameters in the model (Fig 1), such as synaptic conductances, voltage-gated channel densities, length of dendrites, and intensity of incoming stimuli (through a change in the synaptic weights). The default values of parameters that were subjected to change are presented in Table 1. We hypothesised that if unbalanced currents lead to oscillations with a non-zero OM, the change in currents will correspond to a change in OM of a generated signal.

Additionally, to test BSM assumption, we modulated a simulated time course by means of multiplication with an amplitude envelope extracted from real EEG data: $X(t) = A(t) \, m(f, t)$ where $A(t)$—the alpha amplitude envelope extracted from EEG data with the Hilbert transform, $m(f, t)$—oscillations that were produced by the model with $f = 10 \, Hz$, and $X(t)$—modulated signal. Afterwards, using the modulated oscillations we computed the baseline-shift index. The method for estimating the baseline-shift index is presented in the next subsection.

### Estimation of baseline shifts in EEG data

**Participants.** Analysis of empirical data was performed on Leipzig Study for Mind-Body-Emotion Interactions (LEMON) [42] data set, which contains EEG data from 216 healthy participants of young (25.1 ± 3.1 years, range 20–35 years) and old age (67.6 ± 4.7 years, range 59–77 years). Resting-state EEG was recorded from 62 channels placed according to the 10–20

**Table 1. Default parameters of the HNN model [32].**

| Parameter | Value |
|---|---|
| Morphology of the pyramidal neuron | |
| PN II/III—Apical dendrite length ($\mu m$) | 306.0 |
| PN V—First segment of apical dendrite length ($\mu m$) | 680.0 |
| PN V—Second segment of apical dendrite length ($\mu m$) | 680.0 |
| Biophysics of the pyramidal neuron in layer V | |
| Soma K+ channel density ($S/cm^2$) | 0.01 |
| Soma Na+ channel density ($S/cm^2$) | 0.16 |
| Dendrite K+ channel density ($S/cm^2$) | 0.01 |
| Dendrite Na+ channel density ($S/cm^2$) | 0.14 |
| Rhythmic proximal inputs | |
| PN II/III AMPA weight ($\mu S$) | 0.000054 |
| PN II/III NMDA weight ($\mu S$) | 0.0 |
| PN V AMPA weight ($\mu S$) | 0.000054 |
| PN V NMDA weight ($\mu S$) | 0.0 |
| Rhythmic distal inputs | |
| PN II/III AMPA weight ($\mu S$) | 0.000054 |
| PN II/III NMDA weight ($\mu S$) | 0.0 |
| PN V AMPA weight ($\mu S$) | 0.000054 |
| PN V NMDA weight ($\mu S$) | 0.0 |

PN II/III- pyramidal neuron in layer II/III, PN V- pyramidal neuron in layer V.

extended localization, referenced to FCz and grounded at the sternum. The EEG session was carried with alternating 60-sec blocks of eyes-open and eyes-closed conditions, comprising a total of 16 segments starting from the eyes-closed state. In the current study, we used data from young, from 20 to 25 years, and elderly participants, from 60 to 80 years. The choice of participants from the data set was performed based on the following criteria—no habit of smoking, no drug administration at the time of sampling, no depression (based on the Hamilton Depression scale) or any other current mental condition. Additionally, 12 recordings were removed due to significant noise in several channels or the absence of alpha oscillations in the spectrum. After such exclusion, data from 50 young participants and 40 elderly participants were separated into eyes-open and eyes-closed conditions and BSI estimation was performed as described in the next subsection.

**The baseline-shift index.** For the most part, we applied the method for inferring baseline shifts in empirical electrophysiological recordings as described by [14, 16] with some adjustments. The analysis was performed in MATLAB (R2018b, MathWorks) using the BBCI Toolbox [43]. First, the raw recordings were filtered in broadband (0.1–100 Hz) and downsampled to 1000 Hz. Additionally, we applied a notch filter around the frequency of line noise from 48 to 52 Hz. Noisy channels were removed based on visual inspection and spectra inspection, and bad time segments were cut out manually. After cleaning the data, the mean length of the recording in the eyes-closed condition was 493 seconds, the eyes-open condition was slightly shorter—417 seconds. Data were re-referenced to a common average reference. Second, the spectrum of the signal from all electrodes was computed using Welch's method with 50% overlapping 10-sec windows and a corresponding frequency resolution of 0.1 Hz. Alpha peak was determined as the maximum point in averaged spectrum within an 8–13 Hz frequency range. With the obtained value of peak frequency, oscillatory components of activity in the alpha

band were extracted from broadband recording with multivariate Spatio-Spectral Decomposition (SSD) [44]. SSD extracts components that have maximal power at the desired frequency while simultaneously suppressing power in the neighbourhood range, thus increasing the signal-to-noise ratio. The frequency band near the alpha peak was chosen as ±1 *Hz* from the determined individual peak, and flanking frequencies on both sides are from ±2 to ±3 *Hz* from the corners of the centre frequency. Obtained spatial filters were multiplied by minus one, if necessary, so that the maximal value of spatial filter weights was always positive. Further analysis was based on the first five SSD components, as they have the largest signal-to-noise ratio and in general smooth topography. Additionally, time courses of components were visually inspected for some residual high-amplitude noise, and if needed cleaned manually. Third, each component was band-pass filtered in the alpha range, which is ±2 *Hz* around the alpha peak defined individually for each component, and for another signal ($V_{bs}$) the raw data were low-pass filtered with a cut-off frequency of 3 *Hz*. For that purpose, we utilised the Butterworth filter—order two for the alpha band and order four for low-frequency filtering (to control for the steepness of a filter response curve). The amplitude envelope of the bandpass signal ($V_{alpha}$) was obtained using the Hilbert transform. Fourth, the amplitude envelope of alpha oscillations ($V_{alpha}$) was divided into 20 percentile bins according to the magnitude, with the last bin containing the highest amplitude. The low-frequency signal was sorted using the bins composition from the binning of values of $V_{alpha}$. Eventually, amplitudes in each bin were averaged to obtain a mean value of alpha envelope and corresponding low-frequency amplitude. Fifth, the relation between $V_{alpha}$ and $V_{bs}$ was estimated. In the previous research, the baseline-shift index (BSI) was computed as a slope of linear regression [16]. However, in the current study, as the power was different in groups and conditions, we computed BSI as the Pearson correlation coefficient. Custom Matlab functions for EEG analysis are available on Github github.com/astudenova/bsi_matlab.

As for covariates, we computed the power ratio in the alpha band from the following formula $R_a = \frac{P_a}{P_n}$, where $P_a$ is mean spectral power around alpha peak ±1 *Hz*, and $P_n$ is mean spectral power in the flanking frequencies on both sides from ±2 to ±3 *Hz* from the corners of the peak frequency. The power ratio in the low-frequency band was estimated as $R_{lf} = \frac{P_{lf}}{P_n}$, where $P_{lf}$ is mean spectral power from 0.1 to 3 *Hz*, and $P_n$ is mean spectral power in the range from 0.1 to 7 *Hz*.

**Statistical analysis.**   To detect the difference between groups and conditions in absolute values of BSI, we used the ANOVA model (Python module statsmodels, anova_lm) [45], with age and condition (eyes-closed vs eyes-open) as categorical independent variables, and with the power ratio in the alpha band and the power ratio in the low-frequency band as continuous independent variables. Both age and condition were treated as between-group variables. Despite the fact that the same individual's EEG was analysed, for each condition, we applied a distinct set of SSD filters. Therefore, the time courses after SSD filtering represent different sources or a superposition of different sources between eyes-closed and eyes-open conditions. The power ratio values were significantly skewed, and they were log-transformed for ANOVA testing. To determine how strong is the correlation between power ratio and BSI, we used the Pearson correlation coefficient.

To obtain a robust estimation of the significant BSIs, we applied permutation testing. Before binning, a low-frequency signal was cut at random points into 5 intervals (not exceeding 1.5 minutes), which subsequently were randomly shuffled. The permutation distribution was built from computing BSI between the original alpha-filtered time course and shuffled low-frequency surrogate. The number of permutations was 500. The p-value was retrieved as the number of permuted BSIs that increased the original BSI divided by the total number of

permutations. For the evaluation of significance, the sign of BSI was taken into account. The significance threshold was set to 0.05.

## Simulation of networks with various degrees of spatial synchronisation

To examine possible effects of spatial synchronisation on baseline shifts we simulated a time series of synchronous and asynchronous networks [35]. Modelling was performed in MATLAB (R2018b, MathWorks). Each neuron in the network was modelled as a simple oscillator following the sinusoidal wave according to the formula:

$$\mu(t) = A(t)[A_\alpha \sin(2\pi f_\alpha t + \theta_\alpha) + A_\beta \sin(2\pi f_\beta t + \theta_\beta) + r] \qquad (2)$$

where $A_\alpha$, $A_\beta$ are amplitudes of alpha and beta oscillations, respectively,
$f_\alpha$, $f_\beta$—frequencies of alpha and beta oscillations,
$\theta_\alpha$, $\theta_\beta$—phases of alpha and beta oscillations,
$r$—OM of a signal from one neuron,
$A(t)$—amplitude modulation.

 Parameters in this model were set as follows $A_\alpha$ = 1 *a.u.*, $A_\beta$ = 0.25 *a.u.*, $f_\alpha$ = 10 *Hz*, $f_\beta$ = 20 *Hz*, $t$ = 1 *s*, $r$ = −0.4 *a.u.*. Phases of each neuron were sampled from von Mises distribution with different concentration settings [46], using a MATLAB Toolbox CircStat [47]. The probability density function of a random variable with von Mises distribution complies with equation $\phi(\theta) = \frac{1}{2\pi I_0(\kappa)} e^{\kappa \cos(\theta - \mu_0)}$, where $I_0(\kappa)$—is the modified Bessel function of the first kind and order 0, $\mu_0$—mean value, and $\kappa$—the concentration parameter [46]. Phases of beta were computed as shifted to $\pi/4$ alpha phases to recreate the comb-like shape. Amplitude modulation $A(t)$ for the stimulus-induced oscillatory change was simulated as inverted Gaussian with varying widths of the left and right planks. Signals from each neuron were added together to produce a compound signal $X(t) = \sum_{i=1}^{N} \mu_i(t) + \epsilon$ [35], where $N$ = 30000 and $\epsilon$—noise. The type of noise that's implemented was pink noise, as it exists in real EEG recordings. Inverse Fourier transform method was utilised to create the pink noise [48]. In addition to pink noise, Gaussian noise was also appended to the signal, which represented environmental noise [49]. Here $X(t)$ is a single epoch response. The compound signal thus approximately represents a signal typically recorded with EEG/MEG. Further, to imitate the evoked response experimental procedure, the simulation was repeated for 100 epochs, and each time new phases have been sampled from von Mises distribution. The resulting signal was calculated according to the formula $E(t) = \sum_{j=1}^{K} X_j(t)$, where $K$ = 100—number of epochs.

 To design a synchronous and asynchronous network, we varied the concentration parameter $\kappa$: for synchronous network $\kappa$ = 1, for asynchronous network (in which neurons have significant phase lags) $\kappa = 10^{-6}$. The comparison between the two cases was made based on the waveform of a compound signal $X(t)$ and on the time course of the alpha envelope. For the extraction of the alpha envelope, the oscillations of each neuron was "filtered" in the alpha band resulting in a signal $\mu_\alpha(t) = A(t)[A_\alpha \sin(2\pi f_\alpha t + \theta_\alpha)]$, which then was summed across all the neurons in the network $X_\alpha(t) = \sum_{i=1}^{N} \mu_\alpha^i(t)$ and Hilbert-transformed to extract the amplitude envelope over a single epoch.

## Results

### Asymmetric currents lead to non-zero mean oscillations

The simulations with Human Neocortical Neurosolver (HNN) [32] demonstrated that a biologically plausible model of a population of neurons in the neocortex generated alpha frequency oscillations centred around a non-zero oscillatory mean (OM). In HNN, alpha

oscillations are generated by rhythmic inputs (proximal and distal) that arrive in the network in an alternating fashion. The main output of HNN is a current dipole estimate that is computed as the sum of the intracellular currents within the pyramidal neurons' dendrites projected onto a direction parallel to the apical dendrite plane. Note that currents that are flowing down are associated with a negative sign, and upward currents with a positive sign [32]. To provide evidence for non-zero mean alpha oscillations, we systematically varied parameters in the model (Fig 1A) and examined associated changes in OM that were driven by changes in currents (Fig 1B). We simulated alpha oscillations changing only one parameter at a time while holding others to default values (including random seed).

Firstly, we looked into the connection between biophysics and morphology of a pyramidal neuron in relation to the change of OM. Several important currents are modelled in HNN, but we took a closer look at a fast sodium current and a delayed rectifier potassium current, as primary currents responsible for depolarization and subsequent repolarization [33]. Intuitively, when the density of sodium channels on the soma increases, they produce more inward current at the soma level. If all other channel densities are held constant, the increased current would flow along the apical dendrite, creating a more positive OM. On the opposite, an increase of the sodium current in dendrites would cause intracellular currents to flow down the apical dendrite and to the soma, thus forcing OM to shift towards less positive values. Fig 2A indeed confirms the assumption. While the model was generating alpha rhythm with interchanging proximal and distal inputs, an increase in sodium channel densities on the soma led to an increase in backpropagating currents and, subsequently, to an increase in OM. The

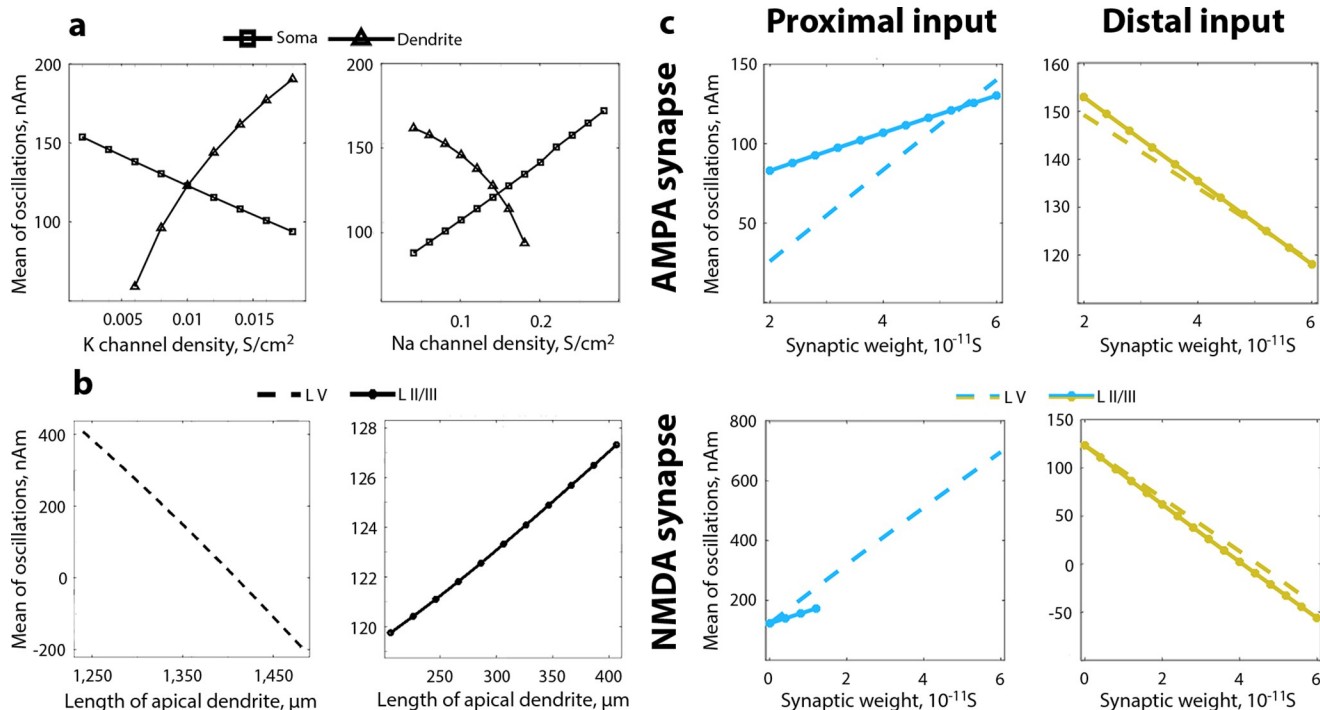

**Fig 2. Relation between biophysics and morphology of a pyramidal neuron and OM. a.** Change in OM of the alpha rhythm with a change in sodium and potassium channel density on soma and dendrites on pyramidal neurons in layer V. **b.** Change in OM with a change of the length of apical dendrite in pyramidal neurons in layer II/III and layer V. Note the y-axis range difference. **c.** Change in OM depending on the strength of incoming inputs realised through excitatory AMPA and NMDA synapses. Only one parameter at a time was changing, all other parameters were kept constant and set by default. OM is computed from 1-second of a simulated signal. L V—layer V, L II/III—layer II/III. The exact default values of parameters that were changed are given in Methods, Table 1.

tendencies for sodium and potassium channels had the opposite directions because sodium ions are moving inside the cell and potassium ions are leaving the cell.

Secondly, since the currents in the apical dendrite contribute primarily to the current dipole change [32], we hypothesised that extending the dendrite would result in a shift of OM. Extending the apical dendrite of pyramidal neurons in layer V indeed caused OM to drop below zero (Fig 2B). Layer V pyramidal neurons are larger than pyramidal neurons in layer II/III, and in particular, basal dendrites on the layer V pyramidal neurons are thicker [37, 50]. Therefore, a current produced by proximal input flows with less resistance through basal dendrites of pyramidal neurons in layer V. When the length of the apical dendrite of pyramidal neurons in layer V is increased this upward current encounters more resistance in the dendrite and thus undergoes a decrease in its strength. For pyramidal neurons in layer II/III, this is not the case since their upward current produced by proximal input is already constrained with thin basal dendrites and an increase in the length of apical dendrite of pyramidal neurons in layer II/III and a subsequent very small increase in OM (Fig 2B) reflect impediments for the current originated from distal input. With these simulations, we demonstrate what has been predicted by previous research [14, 15] that currents in pyramidal neurons are asymmetric and, consequently, elongating apical dendrite produced even more asymmetry.

Thirdly, we varied weights of the AMPA and NMDA synaptic receptors for both proximal and distal inputs (Fig 2C; the ranges for changes are similar as in [38]). The high synaptic weight corresponds to more stimulation coming from a particular input which in turn should be associated with an increase in currents. The proximal drive arrives to basal dendrites of a pyramidal neuron, and currents that are generated in the neuron as a result of proximal stimulation flow upwards (in relation to the cortical surface). Therefore, a bigger stimulation from the proximal sources induced an increase in OM (Fig 2C, left column). The distal drive gives rise to downward currents, and an increase in the distal drive is associated with a decrease in OM (Fig 2C, right column). The tendency was matching for both AMPA and NMDA. AMPA receptors have fast time constants, while NMDA receptors have long time constants [51], therefore, a change in the synaptic weight of NMDA receptors gave rise to a larger change in OM due to a larger effect on the currents. Note that for some cases increase in weights of NMDA receptors causes spiking in the network. For that reason, further change in OM is not presented in Fig 2C (lower panel).

Lastly, as an extreme case, we simulated alpha rhythm with either proximal or distal input. When only the proximal drive was active, a strong current was flowing from basal dendrites to soma and from soma to apical dendrite. This upward current produced a positive current dipole (Fig 3A), with OM being persistently above zero. On the opposite, when only distal input activated the network, currents flew from the top of the apical dendrite in the downward direction (Fig 3C). These dynamics created a shift in OM such that it became less than zero. In both cases, alpha rhythm was preserved. An example of the alpha rhythm simulated with both proximal and distal input using default settings is displayed in Fig 3B. The OM for this case is above zero. The simulations with isolated proximal and isolated distal input represent an exceptional case and serve to demonstrate the effect of proximal and distal drives on dendritic currents and consequently on OM. The other presented results can be conceptualised as intermediate stages between these two extremes.

Owing to the fact that electric and magnetic fields are proportional to primary currents [52], to simulate the amplitude of a signal that is typically observed in macroscopic recordings, oscillations of 200 pyramidal neurons are then multiplied by 300,000 [32] so that 60,000,000 pyramidal neurons give rise to the signal. However, in real settings, the number of synchronously oscillating neurons is rarely constant. Instead, it fluctuates over time [53]. This decrease or increase on the microscopic level is reflected in the changes of the oscillations' amplitude

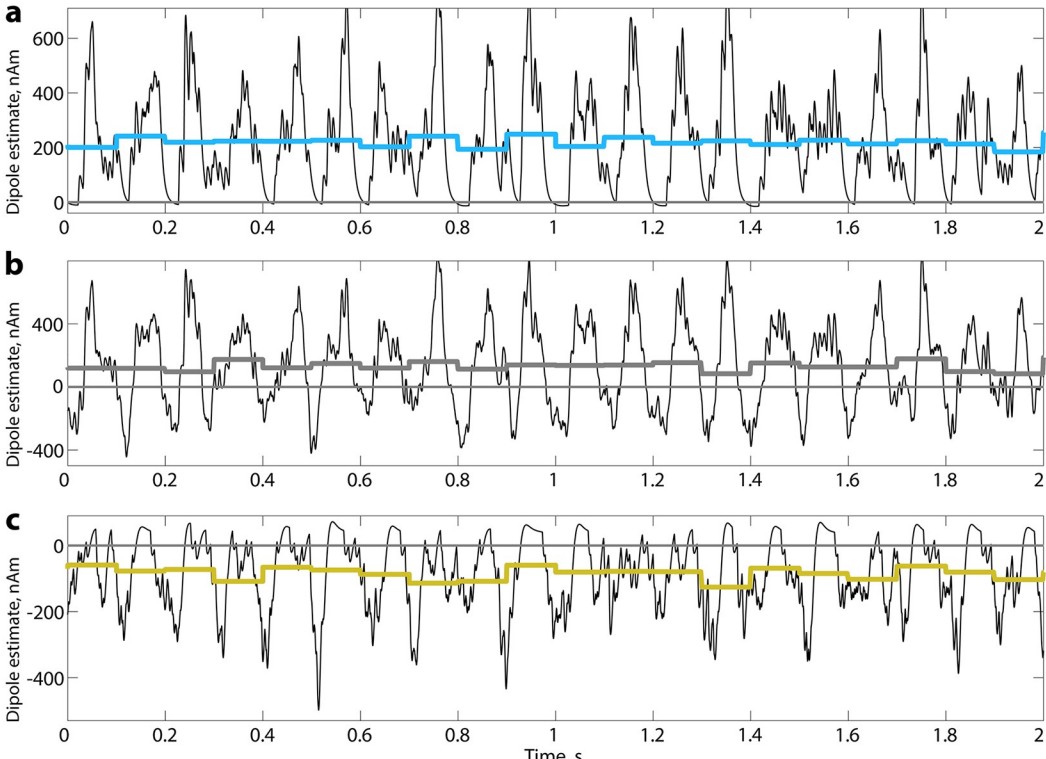

**Fig 3. The alpha rhythm is generated in HNN with alternating rhythmic inputs (proximal and distal). a.** Simulation of alpha rhythm with an isolated proximal input. **b.** Simulation of alpha rhythm with both proximal and distal input using default settings of the HNN model for the generation of the alpha rhythm. **c.** Simulation of alpha rhythm with an isolated distal input. Coloured line—OM that was computed on 100 ms intervals.

[54, 55]. Having that in mind, we extracted the amplitude envelope of the alpha rhythm from real exemplary EEG data and modulated the number of neurons in the model with the extracted envelope. Afterwards, we estimated the baseline-shift index (BSI [16]; see Methods), a measure that allows to detect baseline shifts associated with a non-zero OM in empirical electrophysiological data. As predicted, because OM was positive, the BSI turned out to be positive as well (Fig 4A and 4D). For this simple simulated case, we demonstrate that there exists a straightforward dependence between the amplitude of alpha rhythm and low-frequency amplitude (Fig 4C) that can be measured with BSI. As BSI was previously introduced to quantify baseline shifts associated with alpha oscillations in empirical EEG/MEG data, before applying BSI to a large data set, we additionally juxtaposed its performance in simulations (see S1 Appendix).

## Validating the baseline-shift mechanism on the large EEG data set

Simulated data provide robust and stable oscillations that possess little variation in amplitude and frequency. However, in real data, oscillations tend to occur with highly varying amplitude including periods of no oscillations [56]. That being the case, we investigated the presence of non-zero mean alpha oscillations in a large data set containing resting-state EEG data. We utilised resting-state data because even during rest, the amplitude of alpha oscillations significantly fluctuates over time which in turn should be associated with corresponding baseline

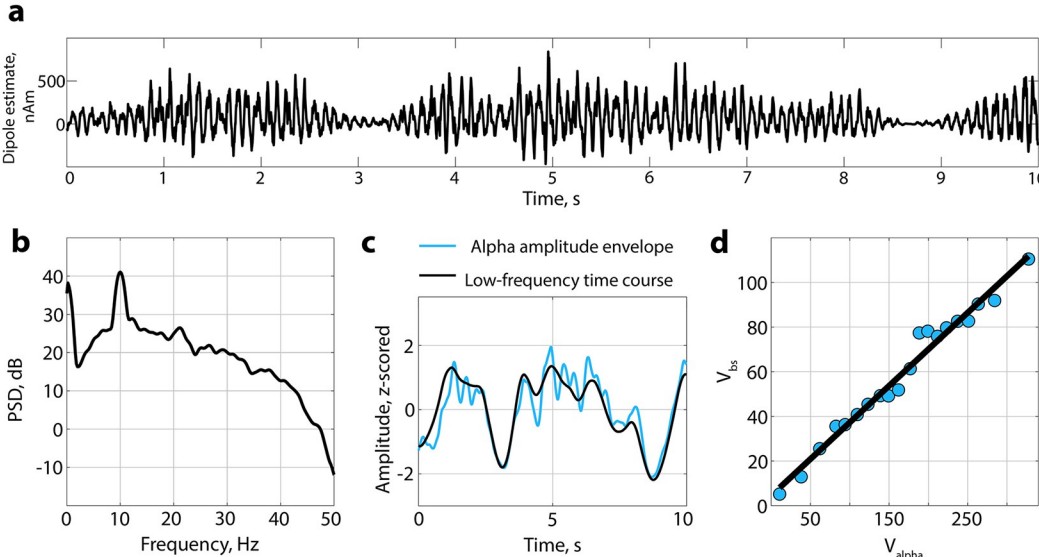

**Fig 4. Alpha modulation from real EEG is superimposed on time series simulated with default settings in HNN. a**. Modulated oscillations. For this case, OM is positive. **b**. A spectrum of the modulated signal. **c**. Correspondence between alpha amplitude envelope and simulated time course filtered in a low-frequency range. Since OM is positive, an increase in alpha amplitude is associated with a baseline shift upwards. **d**. Baseline-shift index (BSI) [16] applied to a simulated and modulated signal. BSI expresses a relation between low-frequency signal and the ongoing rhythm envelope (in the current study, alpha oscillations and associated baseline shifts in the 0.1–3 Hz range). Values of amplitude envelope and low-frequency signal are binned in 20 bins and the relation is estimated as a slope of linear regression or as the Pearson coefficient. The sign of BSI reflects the sign of the underlying OM. For this case, the sign of BSI is positive which is consistent with OM of simulated oscillations. Here, BSI computed as the Pearson coefficient is 0.98.

shifts. To expand the assessment, we analysed data from 90 participants (50 young, range 20–35 years, and 40 elderly, range 60–80 years, participants) with both eyes-closed and eyes-open sessions. From the broadband EEG data, we extracted oscillatory components with Spatio-Spectral Decomposition (SSD) [44], and chose the first five SSD components that typically have the strongest signal-to-noise ratio in the alpha band. The total number of time courses extracted was 900 (500 for young and 400 for elderly participants in two conditions—eyes-closed and eyes-open). Fig 5 shows the representative example of four participants: spatial distribution of the SSD component (Fig 5A), its spectrum (Fig 5B), a baseline-shift index (BSI; Fig 5C), and a corresponding baseline shifts in low frequency associated with the modulation of the alpha rhythm (Fig 5D). For the top two examples, the BSI or the correlation between alpha amplitude envelope and low-frequency amplitude is more than 0.9 (BSI was computed as the Pearson correlation coefficient). For the bottom two examples, the BSI is close to zero. Fig 5 demonstrates how some alpha oscillations have a straight linear relation between their amplitude and amplitude of a low-frequency signal, whereas others do not show this kind of relation. In general, more than 30% of obtained BSIs (39% for young participants, 31% for elderly participants) demonstrated a strong correlation ($|BSI| > 0.7$), and around 30% (31% for young participants, 30% for elderly participants)—moderate correlation ($0.4 < |BSI| < 0.7$; the interpretation of strength of the correlation is based on [57]). With permutation testing (see Methods), the total number of significant BSIs was 50% (53% for young participants, 48% for elderly participants). Critically, 93% of participants had at least one SSD-derived component with a BSI significantly different from zero.

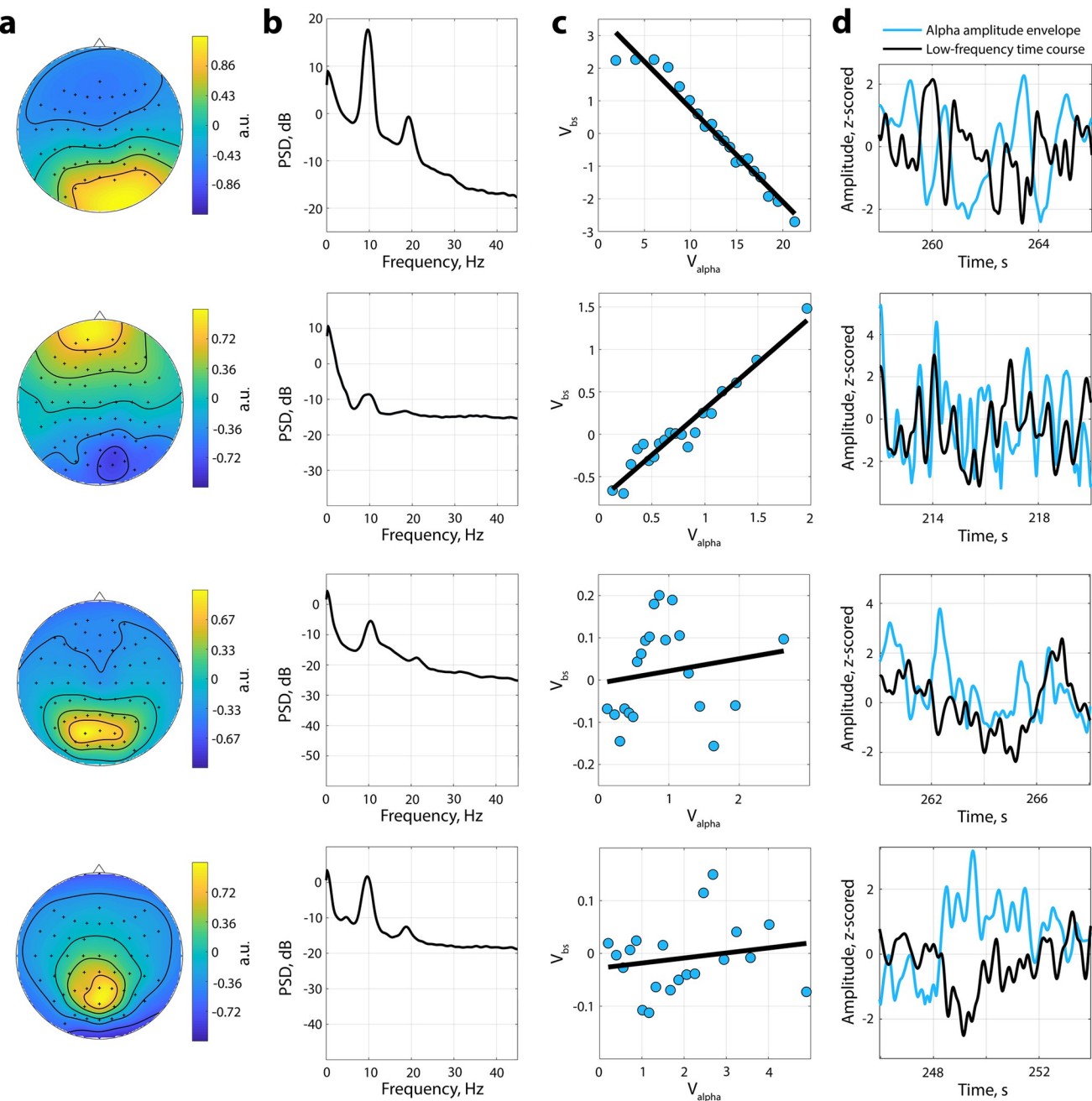

**Fig 5. Baseline shifts in rest EEG data.** Examples of obtained SSD components topographies (Column **a**) and spectra (Column **b**). Column **c**. The association between alpha amplitude envelope ($V_{alpha}$) and low-frequency amplitude ($V_{bs}$)—baseline-shift index (BSI). BSI can be estimated as a slope of linear regression or as a correlation coefficient. For exemplar components, the first row—young group, BSI is negative, the second row, elderly group—BSI is positive, the third row—young group, BSI is close to zero, the fourth row—elderly group, BSI is close to zero. The non-linear relation in the third row may be a manifestation of an interplay between several alpha sources. Column **d**. Correspondence between alpha amplitude envelope and low-frequency signal. If BSI < 0, an increase in the alpha amplitude produces a baseline shift downwards. Contrary, if BSI > 0, an increase in the alpha amplitude produces a baseline shift upwards.

We observed differences in BSI absolute values in young and old participants, in agreement with our hypothesis. Absolute BSI values had a tendency to be larger in the young group (0.564 ± 0.012 in the young group, 0.494 ± 0.015 in the elderly group when averaged across conditions ± standard error of the mean). Fig 6 shows the corresponding data. To test whether

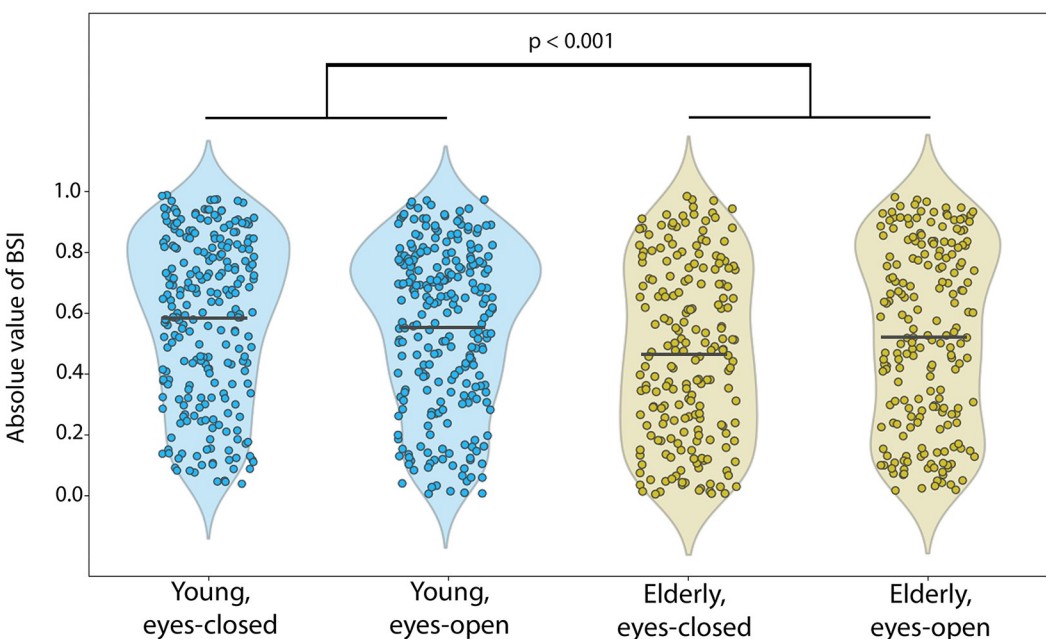

**Fig 6. A difference in the means of absolute values of BSI.** BSI was computed as the Pearson correlation and compared between different groups/conditions. Each dot represents an absolute BSI of one SSD component of each participant. BSI was significantly larger for young participants compared to elderly participants. $p < 0.001$ is based on post hoc Wilcoxon test for age groups when counting BSIs from all SSD components. $p < 0.003$ is based on post hoc Wilcoxon test for age groups when counting BSI averaged over SSD components.

the difference in absolute BSIs was significant between the groups and conditions we applied ANOVA. Additionally, we expected that the power of oscillations may explain the difference between groups. Therefore, we extended the ANOVA model with covariate variables such as power ratio in the alpha band and the power ratio in the low-frequency band that are known to differ in age groups and might have an effect on BSI. In our sample, power ratio in the alpha band was significantly different between the age groups and conditions (ANOVA for age groups $F = 16.62$, $p < 0.001$, $\eta^2 = 0.015$; conditions $F = 194.78$, $p < 0.001$, $\eta^2 = 0.175$; no significant interaction $F = 3.69$, $p = 0.06$, $\eta^2 = 0.003$), as well as power ratio in the low-frequency band (ANOVA for age groups $F = 10.61$, $p < 0.002$, $\eta^2 = 0.011$; conditions $F = 19.41$, $p < 0.001$, $\eta^2 = 0.021$; no significant interaction $F = 0.03$, $p = 0.86$, $\eta^2 = 0.000$). On average, the power ratio in the alpha band was higher in the young group and in the eyes-closed condition. Conversely, the power ratio in the low-frequency band was higher in the elderly group and in the eyes-open condition. Based on ANOVA with absolute BSI as a dependent variable, the difference between age groups was significant ($F = 10.50$, $p < 0.002$, $\eta^2 = 0.011$; post hoc Wilcoxon test for age groups: $p < 0.001$). The difference between eyes-open and eyes-closed conditions failed to reach a significance threshold ($F = 2.64$, $p = 0.10$, $\eta^2 = 0.003$), as well as no interaction in age-condition pairs was detected ($F = 3.17$, $p = 0.08$, $\eta^2 = 0.003$). In addition, in ANOVA, the power ratio in the alpha band and power ratio in the low-frequency band were not strongly associated with the dependent variable (power ratio in the alpha band: $F = 2.28$, $p = 0.13$, $\eta^2 = 0.002$; power ratio in the low-frequency band: $F = 4.68$, $p = 0.03$, $\eta^2 = 0.005$; correlation values between covariates and BSI in age groups and conditions are presented in Fig 7). We also run ANOVA on the mean values of BSI (averaged within subject-condition) and

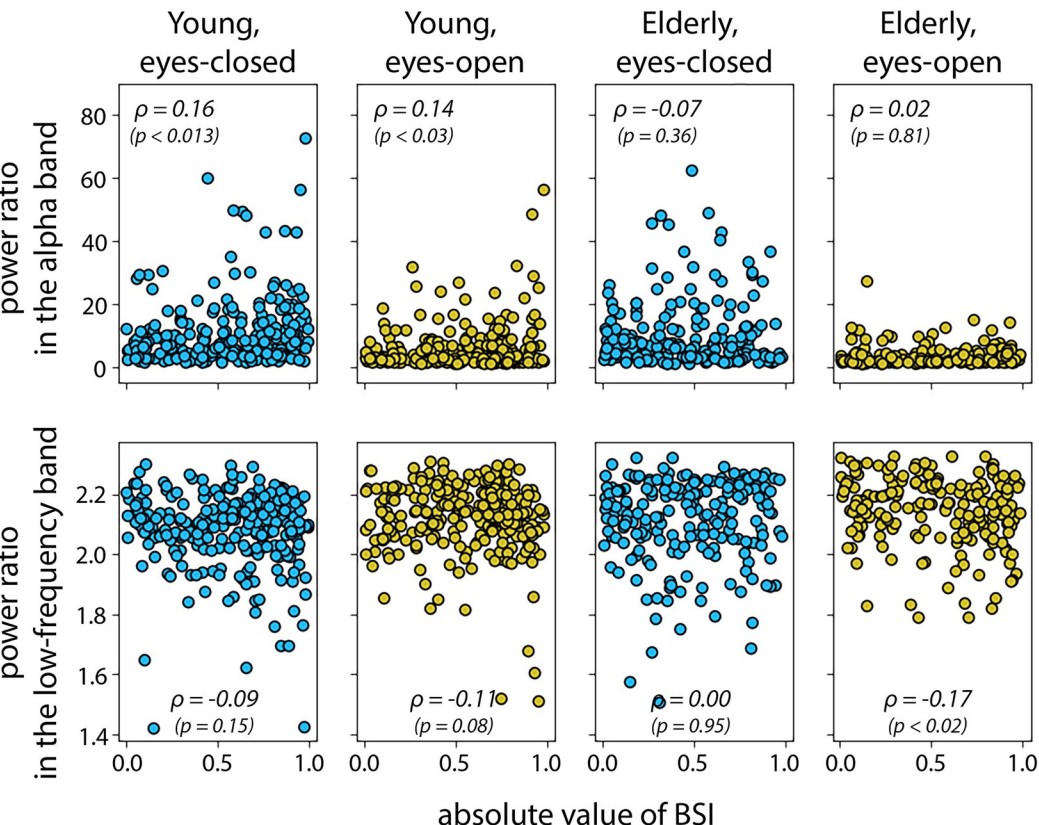

**Fig 7. Correlation between the absolute value of BSI versus power ratio in the alpha band and power ratio in the low-frequency band for pairs of age groups and conditions.** The correlation was computed with the Pearson correlation coefficient. The p-value is presented in the brackets.

obtained similar results—absolute values of BSI were significantly different between age groups ($F = 9.29$, $p < 0.003$, $\eta^2 = 0.049$).

## Effects of spatial synchronisation on the degree of alpha amplitude modulation and baseline shifts

Contrary to our hypothesis, EEG findings indicate that power only weakly correlates with the magnitude of BSI. We observed a significant correlation of BSI and power in the alpha band only for young, but not for elderly participants. In addition to power, the decrease in the synchronisation in the alpha band was reported before for the older population [58]. Therefore, we anticipated that spatial synchronisation among neurons generating oscillations may affect the amplitude of alpha rhythm, and consequently affect the estimation of BSI. Using population modelling [35], we show that the degree of spatial synchronisation has indeed a strong impact on the amplitude of macroscopic alpha oscillations while not having a considerable influence on the generation of the baseline shifts.

For the case of a synchronous network, both alpha modulation and evoked response were present after averaging over epochs (Fig 8, left column). A clear alpha modulation is visible even on a single epoch level. Note that each epoch started with a new phase distribution of individual oscillators, and the central value for distribution was selected randomly (Fig 8,

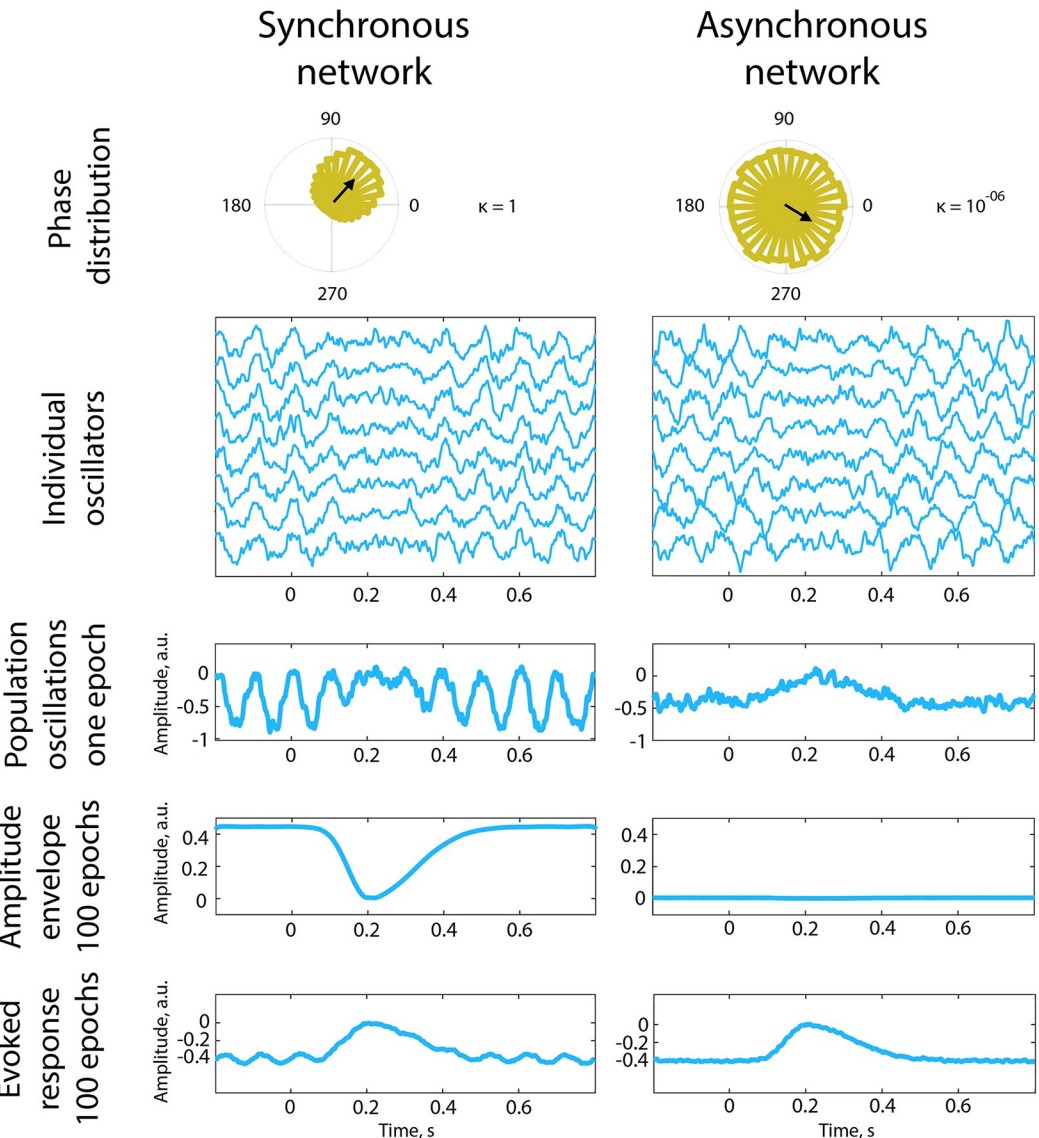

**Fig 8. Synchronous and asynchronous networks.** The synchronous network contained oscillators with small phase delays. Whereas, the asynchronous network had large phase lags between the individual oscillators. At the beginning of the simulation, phases were sampled from von Mises distribution with different concentration settings corresponding to different degrees of within-population synchronisation. $\kappa$ is the concentration parameter that tunes the spread of phases at the beginning of an epoch. The central value of the distribution of starting phases for the displayed epoch is marked by the black arrow. Amplitude modulation was modelled as inverted Gaussian with varying widths of the left and right planks. The network contained 30000 oscillators, the simulation was repeated 100 times to mimic the stimulus-based paradigm. A synchronous network displayed a significantly more pronounced response of the alpha rhythm envelope. For an asynchronous network, the amplitude envelope remained monotonously flat. However, in both cases, individual oscillators underwent the same amplitude modulation. The offset voltage of the evoked response in both cases is negative for two reasons: because we did not perform single-trial baseline correction, and because oscillations were simulated as having a negative mean. In case, when oscillations have a positive mean, non-corrected offset will be positive. After baseline correction, the offset voltage will be shifted to zero.

upper left panel, the central value for the displayed epoch is marked by black arrow). Therefore, alpha oscillations in the evoked response time course (Fig 8, last panel, after averaging of all epochs) were almost cancelled out. Yet baseline shift, representing evoked response, remained since it is not sensitive to the phase of individual oscillators.

For the case of an asynchronous network, alpha modulation was no longer visible even for a single epoch, but the evoked response remained prominent (Fig 8, right column). Visible signs of the modulation of the alpha rhythm (modulation of the amplitude envelope) had mostly disappeared in the time course of one epoch, and even more so in the amplitude envelope averaged over 100 epochs. The attenuation of macroscopic oscillations was due to the phase cancellation among individual oscillators. On the contrary, as simulated oscillators had a non-zero OM of the same magnitude as in the previous case (for the synchronous network), the evoked-response time course replicated the one from the synchronous network simulation, except for the residual alpha oscillations. This is due to the fact that the amplitude of neuronal oscillations critically depends on the phase synchronisation while baseline shifts require only the amplitude modulation of non-zero mean oscillations which is not sensitive to the phase of ongoing oscillations.

## Discussion

A non-zero oscillatory mean (OM) can be conceptualised on a neuronal level as an average value of fluctuations in electrical or magnetic fields produced by neuronal currents. When the ongoing rhythm with underlying non-zero OM is modulated, the baseline shift appears in the frequency range corresponding to a frequency of the amplitude modulation. Therefore, the presence of baseline shifts in real electrophysiological data allows inferring the presence of a non-zero OM. This concept, if valid, has numerous implications for data analysis of EEG/MEG signals and for the interpretation of diverse experimental results.

In the current research, we obtained supporting evidence for the hypothesis of a non-zero OM with the use of computational models and analysis of the real EEG recordings. The Human Neocortical Neurosolver (HNN) [32], the software and the underlying model of a cortical patch was previously validated for human neurophysiological data, including the investigation of mu-rhythm [38]. In our simulations with HNN, we show that pyramidal neurons in layer II/III and layer V of the neocortex generate a current dipole that has a non-zero OM when being integrated over a few oscillations' periods. The non-zero OM was present for diverse settings relating to synaptic conductances, voltage-gated channel densities, and the length of apical dendrites, and the OM variations were consistent with the changes in the distribution of currents inside the neuron. There are other parameters in the model that affect neuronal functioning that we did not mention above. For instance, we did not present results for other channel densities such as channels for $Ca^{2+}$ because their contribution to the generation of a current dipole was small. In fact, the corresponding changes in OM were several orders of magnitude smaller for $Ca^{2+}$ compared to $K^+$ and $Na^+$ currents. Besides, as proximal and distal inputs were driven exclusively by excitatory spiking, we did not present results of other neurotransmitters such as GABA, acetylcholine, noradrenaline etc. In general, for this demonstration, we selected the parameters that, in our view, have the strongest contribution to the primary currents relating to postsynaptic potentials responsible for the generation of alpha oscillations. For almost all settings of tested parameters, OM was non-zero, thus indicating that zero OM should rather be an exception and typically alpha oscillations should primarily be associated with non-zero OM. Although the mean of oscillations is relatively small compared to the amplitude of oscillations, it can still be amplified by the number of synchronously active neurons.

In these simulations, we consider the optimal conditions where the simulated population of neurons is oriented in the same direction, which in turn creates an ideal situation for the detection of asymmetric currents. In the case of a real brain, primary currents of different directions are present due to the intricate folding of the cortex. And such mixing of sources may lead to OM closer to zero. From our simulations, it also follows that fluctuations in currents may on rare occasions lead to close-to-zero mean of oscillations. To further validate the presence of non-zero OM, we also analysed a large data set containing resting-state EEG recordings.

There, we estimated a non-zero OM with a baseline-shift index (BSI) [16], which in this study was computed as the Pearson correlation coefficient. More than 30% of obtained BSIs demonstrated a strong correlation ($|BSI| > 0.7$), and around 30% a moderate correlation ($0.4 < |BSI| < 0.7$). Based on permutation testing, 93% of participants had at least one example of alpha oscillations with BSI that was significantly different from zero, which in general suggests the ubiquity of the non-zero OM phenomenon. When comparing differences in average values of absolute BSIs in age groups and conditions we found that elderly participants had smaller BSIs. This could have been explained by the fact that alpha power decreases with age [26, 31], which was also the case for our sample, and high power may lead to a more robust estimation of BSI due to higher signal-to-noise ratio. We controlled for power in the ANOVA model (with power ratio in the alpha band), but significant difference between age groups remained. Possibly for the elderly group, spatial filtering with SSD resulted in extracted components that represented a larger mixture of alpha sources with varying in sign baseline shifts compared to younger participants. Additionally, the power in low frequency was higher in the elderly group (a finding that some researchers associate with both healthy and pathological aging [59]). Since BSI takes into consideration both alpha oscillations and the low-frequency signal, the noise in the low-frequency range may affect the estimation of BSI thus making the detection of an association between alpha oscillations and baseline shifts difficult for elderly participants.

In our analysis of real EEG, we observed that the correlation between the relative power of alpha and BSI magnitude was generally positive, but not strong. Clearly, it means that the absence of observable macroscopic oscillations may obscure the evidence for BSM. However, it also indicates that there should be other factors, related or not related to oscillations, that can explain a discrepancy between the fluctuation of amplitude envelopes and corresponding baseline shifts. We hypothesised that one of the factors related to oscillations can be the strength of spatial synchronisation.

Therefore, we tested such spatial synchronisation assumption in a population model of simple oscillators simulated as sinusoids with a non-zero offset. In the case of a strong spatial synchronisation, alpha oscillations with non-zero OM demonstrate both pronounced amplitude modulation and corresponding baseline-shift dynamics. On the contrary, reduced spatial synchronisation provided a case when a low-frequency baseline shift was still present, but no amplitude modulation was detected. Subdural recordings research in humans [60] and animal studies [61] demonstrated a rapid decrease in the coherence between neuronal columns with a distance of just a few millimetres. This in turn indicates that oscillatory sources can be active simultaneously but without considerable spatial phase synchronisation thus leading to the attenuation of macroscopic oscillatory amplitude because of the phase cancellations. Such attenuation of amplitude makes it also hard to detect amplitude modulation. Yet baseline shifts are insensitive to the phase alignment of the individual oscillatory sources and therefore the change in the temporal dynamics of the baseline shifts is unaffected by the spatial synchronisation. As a consequence, the prediction of a baseline-shift mechanism of evoked response generation (BSM) that alpha modulation always accompanies evoked response may not be easily observed in empirical EEG/MEG recordings. Importantly, in our simulations, even for the low spatial synchronisation, amplitude modulation was noticeable in the time course of one

oscillator. This kind of data is unavailable with extracranial recordings. Intracranial single- or multi-unit methods may provide a missing link between neuronal level and macroscopic observations, especially for the case of asynchronous networks.

These simulations demonstrated that in addition to a case when both evoked response and alpha amplitude modulation are observed, it is also possible to observe evoked responses due to BSM without detectable amplitude modulation of oscillations (which are, however, present at the level of individual neurons). However, it is an unlikely scenario that the amplitude modulation is present yet no evoked response is detected (apart from the trivial case when neuronal oscillations have a zero OM).

BSM provides far-reaching implications for other phenomena that are investigated in neuroscience. For instance, to a certain degree, phase-amplitude coupling might be a reflection of the modulation of high-frequency activity (with non-zero OM) in a low-frequency range. In this scenario, if the amplitude of alpha oscillations is modulated in a delta frequency range, the phase of baseline shifts in the delta rhythm would be coupled to the increases/decreases in the amplitude of alpha oscillations. Critically, it means that low-frequency oscillations may carry relevant information about a high-frequency activity (here we use high-frequency and low-frequency as relative terms, not predetermined ranges), and thus both phenomena should be considered jointly. For instance, several studies pointed to the role of ultra-slow oscillations in sensory processing [62, 63], but the association between ultra-slow oscillations and alpha rhythm was not explored. However, possibly, those correlations partially stemmed from a non-zero mean theta or alpha oscillations which, when modulated, induced baseline shifts in the slow activity. Importantly, this effect may relate to other frequency bands. Alpha-gamma phase-amplitude coupling is one of the incidences of cross-frequency coupling [64] that may also partially result from the presence of gamma baseline shifts.

Although so far, we were discussing only alpha oscillations, it is plausible that beta- and gamma-band activity also fluctuates around a non-zero OM. There is evidence of high beta and gamma that coincide with early visual evoked responses [65], and even more so, the modulation of a non-zero mean gamma can potentially underlie early evoked responses generation [15].

Many previous studies provided evidence that late evoked response components, for instance, P300, contingent negative variation (CNV), N400, earlier left anterior negativity (ELAN) and others, occur concurrently with the changes in alpha rhythm amplitude [66–71]. Additionally, movement related responses such as readiness potential, also coincide with a decrease in the power of the alpha rhythm [72]. Other research teams assessed cognitive functioning with the simultaneous evoked response and event-related desynchronisation (ERD), with the results showing consistent correlation of evoked response amplitude and/or latency with the corresponding parameters of ERD in the alpha band [19–23]. All these findings can be to a certain extent explained on the basis of the results of our study.

Several previous studies observed evidence that was contrary to the hypothesis that alpha oscillations and evoked responses are manifestations of the same underlying process [24, 25]. However, based on our results, we believe that there are at least two potential reasons for their findings. Firstly, in our study, we applied spatial filtering with SSD to extract alpha oscillations with a high signal-to-noise ratio. Still, even spatially filtered components did not always demonstrate a non-zero mean property. We assume that this might have happened due to source mixing (both in the alpha band and in a low-frequency band) or individual anatomical differences in dipole locations. Consequently, it may be that Fukuda et al. [24] in their study have observed an interplay of several alpha sources, from which only one was related to the evoked response. Secondly, the elderly and clinical populations may have alterations in oscillatory patterns, and Xia et al. [25] recruited elderly participants and elderly patients with cognitive

impairment. In our study, we found that the prominence of baseline shifts was on average lower in elderly participants. However, we observed that the correlation between the relative power of alpha and BSI magnitude was not strong. This indicates that there may be other influences that can explain a dissociation between the fluctuation of amplitude envelopes and corresponding baseline shifts. We hypothesised that one of such influential factors can be the strength of spatial synchronisation and provided comprehensive simulations for this (see Results/Effects of spatial synchronisation on the degree of alpha amplitude modulation and baseline shifts). Therefore, we believe that, in the study by Xia et al. [25], the relation between alpha oscillations and evoked response was concealed due to changes in brain dynamics in the aged or/and diseased population.

Importantly, BSM can coexist with the additive mechanism where evoked responses are generated by the activation of neurons without the concurrent changes in oscillatory dynamics, and with the phase reset mechanism where after stimulus presentation phases of oscillations are shifted and become aligned. However, BSM may explain not only the evoked response per se but also the change in amplitude of evoked response, when it is generated due to a combination of mechanisms.

The current study has several limitations that could be potentially addressed in future studies. In the analysis of EEG data, we used spatial filtering with SSD which helped to retrieve alpha oscillations with a high signal-to-noise ratio. Yet these components can still reflect a mixture of oscillations with varying directions of baseline shifts. Moreover, we did not set any specific hypothesis about the location or function of the alpha rhythm with a non-zero mean. Instead, we explored all available SSD components containing alpha oscillations and verified their agreement with BSM. On the contrary, if the research question involves a certain evoked response such as P300, source reconstruction would be necessary. If BSM is to be tested for a particular evoked response, one should examine oscillations that may be associated with this response in the source space. Furthermore, the question of frequency tuning remains unanswered. For the computation of BSI, we bandpass-filtered the broadband data around the individual alpha peak. However, for low frequency, we used the same predetermined range from 0.1 Hz to 3 Hz. Possibly, the range of alpha modulation may differ in participants and even within one participant for different alpha sources. In the light of recent studies that show how alpha frequency ranges may differ across the population, even with speculations that for some participants alpha oscillations may lie outside the typical alpha range [73], we assume that more careful frequency tuning may be beneficial for the detection of baseline shifts.

In the current study, we observed that amplitude modulation of the alpha oscillations is reflected in the low-frequency component as predicted by BSM, which is the evidence for the existence of a non-zero OM in alpha oscillations. This, in turn, provides further support for the idea that certain evoked responses and neuronal oscillations may share common neurophysiological mechanisms. Our results thus indicate that a number of motor and cognitive responses which coincide in time with the modulation of alpha oscillations may be interpreted in the context of oscillatory neuronal dynamics.

## Supporting information

**S1 Appendix. BSI performance on simulated data with 1/f noise.** In the following appendix, we provide additional simulations to show that the estimation of BSI is not biased when the oscillatory signal has a 1/f spectrum.
(PDF)

## Author Contributions

**Conceptualization:** Arno Villringer, Vadim V. Nikulin.

**Formal analysis:** Alina A. Studenova.

**Investigation:** Alina A. Studenova.

**Project administration:** Vadim V. Nikulin.

**Resources:** Vadim V. Nikulin.

**Software:** Alina A. Studenova.

**Supervision:** Vadim V. Nikulin.

**Writing – original draft:** Alina A. Studenova, Vadim V. Nikulin.

**Writing – review & editing:** Alina A. Studenova, Arno Villringer, Vadim V. Nikulin.

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
