## [Decision Letter · Decision Letter 0]

8 Mar 2022

Dear Studenova,

Thank you very much for submitting your manuscript "Non-zero Mean Alpha Oscillations are Evident in Computational Model and Empirical Data" for consideration at PLOS Computational Biology.

As with all papers reviewed by the journal, your manuscript was reviewed by members of the editorial board and by several independent reviewers. The overall impression is of a good and solid paper. Some organizational and methodological aspects still need some attention.

In light of the reviews (below this email), we would like to invite the resubmission of a revised version that takes into account the reviewers' comments.

We cannot make any decision about publication until we have seen the revised manuscript and your response to the reviewers' comments. Your revised manuscript is also likely to be sent to reviewers for further evaluation.

Sincerely,

Daniele Marinazzo

Deputy Editor

PLOS Computational Biology

Daniele Marinazzo

Deputy Editor

PLOS Computational Biology

Reviewer's Responses to Questions

**Comments to the Authors:**

Reviewer #1: In this paper Studenova et al investigate whether neural oscillations and evoked potentials could (at least in some cases) reflect the same underlying phenomena. In particular, they investigate the idea that neural oscillations may have a non-zero mean, and that this, paired with amplitude modulation in response to stimuli, can plausibly explain evoked potentials. To investigate this, they use both simulations, using the Human Neocortical Neurosolver tool, and empirical data analysis, measuring the baseline-shift index, a putative marker of non-zero mean oscillations, in EEG data from young and old participants. They report that both the modelling and empirical analyses are consistent with non-zero mean oscillations, in support of the baseline-shift mechanism of evoked potentials.

Overall, I find this paper to be an interesting and useful contribution to conceptualizing and investigating features of neural data, and I think the main analyses are sensible and compelling. I particularly like the combination of simulation and empirical work, and in particular the use of modelling that addresses the potential underlying physiology. Overall, I think this paper makes a useful contribution to the literature. I have some relatively minor comments and suggestions for the manuscript, in terms of a question about the methods and some notes on the discussion of the findings.

I feel that, at times, the paper and results don’t clearly differentiate if the findings support the potential contribution of the baseline-shift mechanism (with other potential mechanisms still in play) or whether they suggest that baseline-shifts ubiquitously explain evoked potentials. For example, the final sentence of the abstract states that “Overall, our results provide strong support for the unification of neuronal oscillations and evoked responses.” Where I find the paper convincing in arguing that there exist non-zero mean oscillations, and that these plausibly at least contribute to evoked-potentials, to claim there is a convincing demonstration that the oscillations and evoked potentials can be completely merged, as is implied here, is not demonstrated. The paper itself acknowledges this in other places, stating “It should be noted that these mechanisms are not assumed to be mutually exclusive and may co-exist” in the introduction. I think the abstract should be careful not to overstate the overall conclusion, and this line should be edited.

The results are also presented in a way to somewhat over-emphasizes the subset of cases in which the baseline-high shift index is particularly high, first discussing, and only showing examples of high correlation, and not explicitly noting the number of cases that have a low correlation. This is despite the average baseline-shift of ~0.5 being a more modest finding, which is only noted a bit later in the results. It seems that many baseline-shift scores are close to zero is more consistent with baseline shift contributing / sometimes happening, rather being a clearly ubiquitous phenomenon. I think the results section on the empirical data could do better to acknowledge the variance in the findings. For example, instead of 3 selected and non-representative examples in Figure 3 (that are all quite similar), it may be useful to instead / also show examples from each group that show an example of the average baseline shift index – this would be more reflective of the pattern of results and help to visualize more common scenarios. Highlighting this variance could also surface a discussion of potential explanations of this variation. For example, it could be noted that this variance may be indicative of the extent to which other mechanisms may be at play.

Methodologically, the baseline shift is broadly a correlation between different frequency ranges in the data, computed from filtered traces. I wonder what the “null model” is for such a measure. For example, for white noise, this measure should, on average, be zero. However, neural data is not white noise, and the pink noise “background” of neural data implies that this measure would be expected to have some non-zero correlation, even absent of any oscillations. This seems to imply that one could see non-zero BSI values given the nature of the data, but absent of any non-zero mean oscillations. Has this measure been characterized in such cases to estimate the influence of colored noise signals on the data? Given the use of SSD, we can expect that the these components do contain oscillatory activity, but given the presence of 1/f activity (visible in Figure 5, for example), it would be useful to know the expected value of the BSI measure under an appropriate null model, to evaluate if and when the measured values are over and above this value. To the extent that this is not known, it may be important to at least acknowledge this potential issue. Conceptually, this issue is similar to investigations showing that, for example, that measures computed across different frequency ranges can be conflated by the overall 1/f activity of the data.

Related reference:

Donoghue, T., Dominguez, J., & Voytek, B. (2020). Electrophysiological frequency band ratio measures conflate periodic and aperiodic neural activity. Eneuro, 7(6).

Finally, I think the paper could benefit from some further discussions of the implications of the findings:

- In the introduction, the paper mentions some with results that are inconsistent with the baseline-shift mechanism (Fukuda et al, 2015 & Xia et al, 2020), and implies that this paper will address factors that might explain these conflicting results. However, this point is not revisited. How do the current results relate to such findings? It seems that this is consistent with baseline-shift explaining some of the variance of evoked potentials, but other mechanisms being at play, but the relation of this paper to prior reports with different findings is not revisited in the discussion, where it would be useful to do so.

- Based on the demonstrations (for example, Fig. 7), it seems that an implication of baseline shift is that evoked potentials themselves are not expected to be deflections around zero (the computed evoked responses of the simulations having non-zero mean). I presume the implication is that in real data, pre-processing steps such as demeaning or high-pass filtering may obfuscate this. Do these findings imply anything different that could be done in the analysis of evoked potentials to further explore the baseline-shift mechanism?

- Alpha oscillations are not continuously present across time and space. Relatedly this paper explicitly excludes patients that do not demonstrate clear alpha oscillations, which seems to be a potential issue give the goals of this analysis. Is the variability of oscillations a potential challenge to the baseline-shift proposal? What is the implication for subjects in which we do not observe clear oscillations?

I understand that each of these points could be entire investigations themselves (and may be discussed in prior work that could be referred to), such that there may be minimal space to fully address these points, but given the potentially broad implications of this paper, I think it’s worth trying to explore some of the points.

Figure notes:

- In Figure 5, in the power spectra, the steep drop off around 50 Hz is presumably due to the notch filter applied to the line noise. Since this drop off is irrelevant to the analyses, the spectra could be limited to a lower frequency to avoid visualizing this drop, as it’s a bit distracting.

- In Figure 5, it might be useful to indicate which groups the examples come from.

Reviewer #2: This paper examines the biophysical bases of non-zero mean alpha oscillations in electrophysiological data using realistic biophysical models and also a reanalysis of large EEG dataset. Results from simulations are quite interesting and worthy of publication. However, the paper is not as well organized. The introduction does not set up the questions, rationale and hypotheses being tested well. The reanalysis of large EEG data is not well justified. The final analysis of the role of spatial synchronization of alpha oscillators and how that could lead to differential empirical findings is not well described either. But the overall approach is noteworthy.

Title is also a bit misleading and need context. What does it mean to be evident in empirical data?

Abstract - The abstract is currently a bit too sparse. The ideas here extend also to iEEG and ECoG. A bit more needs to be said about the biophysical model and what parameter ranges were explored. What are participants in a model? What is the complementary neuronal-ensemble modeling? What discrepancy is being modeled and what is the explanation? What alternative hypothesis is ruled out?

Introduction - Distinction between phase-locked vs non-phase-locked task-induced neural oscillation should be made clear upfront.

It is very strange to say "...there is no clear cut evidence in strong support of either of mechanisms." when the authors have cited so many papers for each type of mechanisms for the evoked response.

Seems redundant to say "The third mechanism - BSM - is based on two prerequisites: 1) sensory stimuli or movements should modulate the amplitude of ongoing oscillations (these ndings being conrmed in almost all EEG/MEG studies on oscillations); 2) neuronal oscillations have a non-zero OM. ..." when this point has just been stated in the previous paragraph. The subsequent first two predictions are also a repetition of the same points. Isnt the third prediction mentioned in this paragraph just a restatement of the first one that - if evoked responses are a result of post-stimulus induced amplitude modulations their timing and spatial distribution would be similar? A bit more need to be said about the two studies that did not find evidence for such correlation. Actually, it is unclear that is what was found in the Fukuda et al. study. They suggested that both amplitude modulation and evoked responses independently contribute to working memory effects rather than being a unitary process but could that be something unique to memory encoding?

"On a membrane level ... placed asymmetrically on a neuron ..." What does asymmetry refer to here? On a cellular level how does orientation contribute to asymmetry? Finally on the exogenous inputs level, again it is unclear what is symmetric and what is being referred to as asymmetry here. Overall this paragraph needs to be made clear. The last two sentences are also confusion because there is a suggestion that some of this is already known, if so what is new that is being investigated here.

The paragraph starting with "Despite that ..." refers to previous studies but has no citation. The point of this paragraph is unclear.

There needs to be a clearer motivation for the use of the HNN. What exactly is the validation in a larger dataset? What is new here? What is actually being done here? The final goal of the study is quite vague. What are some hypotheses that are being tested here? Overall, the goals and hypotheses need to be more clearly laid out here.

Results - the color scheme in figure 1b needs to be explained more clearly. What is the main focus of this figure? A bit unclear if the frequency of oscillations generated in figure 2 are due to the frequency of the proximal or distal input drive?

WIth the intro suggesting membrane, cellular and network level properties as contributors to non-zero OM it may make more sense to present the results in a parallel structure. It seems like there is no zero-mean in the model across a range of parameters. Are they any conditions where they will be zero-mean? There seems to be a clue in figure 3B but nowhere else. The varying axes in figure 3 is also confusing but interesting. Is the DC levels expected to be much smaller than observed oscillatory amplitudes? That seems to be the case in some of the examples but not in all. It may help to look at the relative alpha power and the OM together.

A bit more needs to be said about how figure 4d is computed from the other parts of this figure. I think a figure showing how the BSI is computed will be helpful in this figure. The meaning of the BSI slope should also be better explained.

"The sign of BSI is positive which is consistent with OM of simulated oscillation. BSI " What does this mean?

Figure 5 analysis is not well motivated. There is no figure on the correlations between BSI and low-frequency etc. What is the goal of the ANOVA described in page 8? What hypotheses are actually being tested here and why?

The transition from simulations to real data is abrupt.

The section on effects of spatial synchronization on alpha amplitude modulation and baseline shifts is also quite poorly motivated. Figure 7 is poorly explained.

Much of the discussion seems to be a restatement of many of the results and the broader implications are somewhat missing, as are alternative interpretations and confounds from prior studies etc.

**Have the authors made all data and (if applicable) computational code underlying the findings in their manuscript fully available?**

Reviewer #1: Yes

Reviewer #2: Yes

PLOS authors have the option to publish the peer review history of their article (what does this mean?). If published, this will include your full peer review and any attached files.

Reviewer #1: **Yes: **Thomas Donoghue

Reviewer #2: **Yes: **Srikantan Nagarajan
---

## [Decision Letter · Decision Letter 1]

1 Jun 2022

Dear Studenova,

We are pleased to inform you that your manuscript 'Non-zero Mean Alpha Oscillations Revealed with Computational Model and Empirical Data' has been provisionally accepted for publication in PLOS Computational Biology.

Best regards,

Daniele Marinazzo

Deputy Editor

PLOS Computational Biology

Daniele Marinazzo

Deputy Editor

PLOS Computational Biology

Reviewer's Responses to Questions

**Comments to the Authors:**

Reviewer #1: I commend the authors on a thorough and thoughtful response and revision to the manuscript. My original comments and concerns have all been adequately addressed. I think this paper provides an interesting and productive contribution to the literature.

Reviewer #2: The authors are to be applauded in their extensive revisions and for clarifying all questions raised by both reviewers. They have comprehensively addressed all prior criticisms satisfactorily and substantially improved the paper. This will be a nice contribution to the literature in terms of describing mechanisms of non-zero oscillatory means.

**Have the authors made all data and (if applicable) computational code underlying the findings in their manuscript fully available?**

Reviewer #1: None

Reviewer #2: None

PLOS authors have the option to publish the peer review history of their article (what does this mean?). If published, this will include your full peer review and any attached files.

Reviewer #1: **Yes: **Thomas Donoghue

Reviewer #2: No

---

## [Editor Report · Acceptance letter]

20 Jun 2022

PCOMPBIOL-D-22-00143R1 

Non-zero Mean Alpha Oscillations Revealed with Computational Model and Empirical Data

Dear Dr Studenova,

I am pleased to inform you that your manuscript has been formally accepted for publication in PLOS Computational Biology. Your manuscript is now with our production department and you will be notified of the publication date in due course.

With kind regards,

Zsofia Freund
